# Phagocytosis Checkpoints in Glioblastoma: CD47 and Beyond

**Amber Afzal** [1], **Zobia Afzal** [1], **Sophia Bizink** [1], **Amanda Davis** [1], **Sara Makahleh** [1], **Yara Mohamed** [1] and **Salvatore J. Coniglio** [1,2,*]

1   School of Integrative Science and Technology, Kean University, Union, NJ 07083, USA;
    afzalam@kean.edu (A.A.); afzalz@kean.edu (Z.A.); bizinks@kean.edu (S.B.); davamand@kean.edu (A.D.);
    makahles@kean.edu (S.M.); mohameya@kean.edu (Y.M.)
2   Department of Biological Sciences, Kean University, Union, NJ 07083, USA
*   Correspondence: coniglsa@kean.edu

**Abstract:** Glioblastoma multiforme (GBM) is one of the deadliest human cancers with very limited treatment options available. The malignant behavior of GBM is manifested in a tumor which is highly invasive, resistant to standard cytotoxic chemotherapy, and strongly immunosuppressive. Immune checkpoint inhibitors have recently been introduced in the clinic and have yielded promising results in certain cancers. GBM, however, is largely refractory to these treatments. The immune checkpoint CD47 has recently gained attention as a potential target for intervention as it conveys a "don't eat me" signal to tumor-associated macrophages (TAMs) via the inhibitory SIRP alpha protein. In preclinical models, the administration of anti-CD47 monoclonal antibodies has shown impressive results with GBM and other tumor models. Several well-characterized oncogenic pathways have recently been shown to regulate CD47 expression in GBM cells and glioma stem cells (GSCs) including Epidermal Growth Factor Receptor (EGFR) beta catenin. Other macrophage pathways involved in regulating phagocytosis including TREM2 and glycan binding proteins are discussed as well. Finally, chimeric antigen receptor macrophages (CAR-Ms) could be leveraged for greatly enhancing the phagocytosis of GBM and repolarization of the microenvironment in general. Here, we comprehensively review the mechanisms that regulate the macrophage phagocytosis of GBM cells.

**Keywords:** glioblastoma; glioma; macrophage; phagocytosis; CD47; CSF-1R; siglec

## 1. Introduction

### 1.1. Glioblastoma and Immunotherapy

Glioblastoma multiforme (GBM) is the most common brain cancer with an incidence of 3.26 cases per 100,000 annually. GBM is an almost invariably fatal brain cancer with a median survival from time of prognosis of approximately 14 months [1]. Hallmarks of GBM include a high degree of invasiveness, frequent recurrence, and resistance to chemotherapy. There are multiple subtypes of GBM, each with a distinct molecular signature [2]. The standard of care involves surgical resection and irradiation (IR) with temozolomide (TMZ) therapy. This treatment regimen, however, is largely ineffective and has considerable side effects. New therapies are urgently needed. Immunotherapy involves harnessing the immune system to mediate the destruction of cancer cells. The human immune system relies on a balance of activating versus inhibitory signals which ensures an appropriate response to infectious pathogens and an avoidance of overstimulation and potential tissue damage. Immune checkpoints (ICs) have been identified in several immune subtypes and mainly function to dampen the adaptive immune response. These ICs include the cell surface receptors cytotoxic T-lymphocyte antigen-4 (CTLA-4) and programmed cell death protein 1/programmed cell death-ligand 1 (PD-1/PD-L1). Immune checkpoint inhibitors (ICIs) have been developed to release these immunological "brakes" on the cytotoxic T-cell arm against tumors. Monoclonal antibodies against CTLA-4 (Ipilimumab and tremelimumab), PD-1 (pembrolizumab, nivolumab, cemiplimab, and dostarlimab),

and PD-L1 (durvalumab, avelumab, and atezolizumab) have been developed and are currently used in the clinic to treat a variety of cancers. These ICIs have demonstrated remarkable efficacy in a subset of patients [3]. Unfortunately, their efficacy in treating GBM thus far has been very disappointing [4,5]. Several large-scale clinical trials using ICIs have been carried out in GBM patients. In the CheckMate 143 phase III clinical trial, nivolumab was compared with the anti-VEGF monoclonal antibody bevacizumab in patients with recurrent GBM [6,7]. Two other recent phase III clinical trials tested the ability of nivolumab to enhance standard-of-care therapy [8–10]. The CheckMate 498 trial used nivolumab with IR versus TMZ + IR and CheckMate548 used nivolumab with IR/TMZ. In all of these studies, no benefit was observed in the patients administered nivolumab as either monotherapy or in combination therapy. It is noted that it may be necessary to screen for high PD-L1 expression to identify a subset of patients who are more likely to respond. Other smaller-scale trials have recently been carried out. Of particular note is a phase II study by Cloughesy et al. where pembrolizumab was used as a neoadjuvant [11]. Patients who received pembrolizumab two weeks before surgery had a statistically higher median overall survival. Another study using nivolumab as a neoadjuvant observed higher T-cell infiltration and clonal diversity, although no patient effect on survival was measured [12]. The ineffectiveness of ICIs in treating GBM is likely due to several factors. The brain is an immunoprivileged organ which limits access to certain immune cells such as CTLs. In addition, GBM tumors are also notoriously immunologically "cold" and effector cells that manage to traffic to the tumor are unlikely to mount a sufficient response [4,5,13]. Recently, immune checkpoints involved in tumor–macrophage interactions have gained attention. Macrophages are likely one of the first immune cell types that cancer cells encounter and these early interactions are likely to dictate the progression of the tumor.

### 1.2. Tumor-Associated Macrophages

It is now widely appreciated that the microenvironment plays a central role in dictating the immune response to the tumor and the overall progression to malignancy [14,15]. The microenvironment consists of many immune cell types including lymphocytes and macrophages (TAMs), the latter being especially prevalent in GBM. Approximately 30–50% of the GBM tumor mass is comprised of macrophages, microglia, and other cells of the myeloid lineage [16–20]. Labeling experiments have shown that most are derived from peripheral blood, although it is likely brain resident microglia also play an important and non-redundant role in tumor progression [21,22]. Extensive paracrine signaling occurs between glioma cells and TAMs [23–28]. In general, glioma cells secrete factors which recruit and induce TAMs to acquire a pro-tumoral phenotype (often referred to as M2). These reprogrammed TAMs in turn promote glioma cell invasion, chemoresistance, and immunosuppression of the microenvironment. It is recognized that the standard "M1 vs. M2" model of macrophage polarization is an oversimplification and macrophages display phenotypes along a continuum [29,30]. It is generally agreed, however, that M1 macrophages express cytokines such as IL-12 and the cell surface markers CD80 and CD86, whereas M2-polarized macrophages express immunosuppressive cytokines such as IL-10 and TGF-b and cell surface markers such as CD163 and CD206. An attractive strategy is to use therapy to "reprogram" TAMs away from an M2 toward an M1 anti-tumoral phenotype. This is likely to yield substantial benefits as the pro-tumorigenic effects mediated by M2 macrophages including glioma invasion and chemoresistance, and immunosuppression will be reversed and M1-polarized TAMs will have the ability to recruit and activate potent effector cells. These include cells such as NK cells and cytotoxic CD8+ T lymphocytes to attack tumor antigen expressing glioma cells and the release of additional pro-M1 factors which would propagate a positive feedback loop.

There are subtleties in how the tumor interacts with the myeloid compartment of the microenvironment. For example, so-called glioma stem cells (GSCs) are a subset of tumor cells which express a different profile of chemokines, growth factors and interleukins, and other factors from the bulk tumor population [31–35]. As a consequence of this differential cytokine

expression, GSCs are thought to interact in a more specialized way with TAMs. GSCs also express specific cell surface proteins such as CD44, CD133, and nestin which could mediate special interactions with TAMs within the microenvironment [36]. In addition to heterogeneity among the glioma cells, as mentioned previously there are various subpopulations of myeloid cells within the microenvironment including resident brain microglia, macrophages derived from monocytes in the blood, CD11c+ dendritic cells, immature Gr1+ monocytoid cells, and myeloid derived suppressor cells (MDSCs). Each of these cells in turn can have overlapping and distinct effects on tumor progression [18,20,37–42]. Further complicating the investigation of the relative contribution of these macrophage/myeloid subpopulations in GBM progression is that there are substantial differences in the cell-specific markers between mice and humans employed to distinguish between macrophage subsets [43].

### 1.3. Macrophage Phagocytosis of Glioma Cells

Most of the myeloid cell types associated with GBM tumors, including macrophages, microglia, and dendritic cells, retain the capacity to carry out phagocytosis under certain conditions [44–52]. The current paradigm is that tumor cells experiencing oncogenic cellular stress express and present "eat me" signals such as surface exposed calreticulin, High mobility group box 1 protein (HMGB1), and phosphatidyl serine (PS), among others [53,54]. When these signals are detected by tissue macrophages, the result is the rapid and efficient phagocytosis of the tumor cell. However, tumor cells eventually upregulate surface "don't eat me" receptors, the most well-characterized of which is CD47. CD47 is a transmembrane protein containing an extracellular immunoglobulin (Ig) domain which engages SIRP receptors on macrophages that strongly inhibits phagocytic signaling pathways [55]. There are three SIRP receptors in humans, the most studied being SIRP alpha [56]. The cytoplasmic domain of SIRPa contains an immunoreceptor tyrosine-based inhibition motif (ITIM) which recruits SHP1/2 phosphatases involved in downregulating immune signaling, particularly pathways that are involved in phagocytosis [57]. The Weissman lab has pioneered the use of therapeutic anti-CD47 antibodies to treat solid tumors in a variety of mouse models [58–61]. Preclinical studies with anti-CD47 blocking antibodies have been promising in many cancer models, and humanized anti-CD47 monoclonal antibodies such as Hu5F9-G4 (Magrolimab) have been developed [62]. Unfortunately, as is the case with most current immune checkpoint therapies, anti-CD47 seems to be significantly less effective against solid tumors in the clinical trials conducted thus far [63]. In addition, CD47 is expressed at high levels in red blood cells and other cell types which could make it less than ideal as a targeted monotherapy. Anti-SIRP approaches are currently in development to perhaps escape these limitations [64]. Other "don't eat me" surface immune checkpoint molecules include certain sialic acid containing glycoproteins and glycolipids which engage the sialic acid-binding immunoglobulin type lectins (siglec) proteins which inhibit the immune response.

Macrophages are also responsible for clearing apoptotic GBM cells [65–67]. Paradoxically, the pathways that regulate phagocytosis of apoptotic cells has been shown in many contexts to induce an anti-inflammatory response [68]. For example, bone marrow derived macrophage (BMDM) phagocytosis of GBM cells was shown in-vitro to result in lower secretion of IL-1b and TNFa and an increase in anti-inflammatory IL-10 cytokine production [69]. Phagocytosis also resulted in an increase in macrophage expression of PD-L1 and PD-L2. These mechanisms almost certainly evolved to prevent an inappropriate inflammatory response during the clearance of apoptotic cells which occurs during normal physiological processes. It is unclear if inducing apoptosis of tumor cells will augment or inhibit the in vivo antigen presentation of tumor antigens. There has been some work showing alternative forms of cell death can be induced in cancer cells to overcome this immunosuppressive effect. For example, the downregulation of HSP70 resulted in a caspase-independent "apoptosis-like" effect in breast and colorectal tumor cells, whereby phagocytic clearance was not accompanied by immunosuppressive cytokines [70]. Nonetheless, understanding the full spectrum of molecular pathways which govern TAM phagocytosis of GBM cells

will likely result in great strides toward additional checkpoint inhibitor therapy. Here, we review these pathways in a comprehensive fashion. The main points are summarized in Table 1.

**Table 1.** Summary of pathways that regulate TAM phagocytosis of GBM cells.

| Phagocytic Checkpoint Receptor (on Tumor Associated Macrophage) | Ligand (on GBM Cell) | Summary | References |
|---|---|---|---|
| SIRPa | CD47 | CD47 "don't eat me signal" engages SIRPa on TAM and prevents phagocytosis | [71–77] |
| CSF-1R | Membrane CSF-1 | Membrane-bound CSF-1 on GBM cells induces strong phagocytic response in TAMs | [78–80] |
| BACE-1 | Beta-amyloid/unknown | Inhibition of BACE-1 enhances TAM mediated phagocytosis of GBM | [81,82] |
| TREM2 | Unknown | TREM2 correlates with enhanced phagocytosis of GBM | [83] |
| Siglec-H/1/15 | Sialic acid modified cell surface proteins | Inhibition of siglec expression on TAMs enhances phagocytosis | [84–87] |

## 2. Pathways Regulating TAM Phagocytosis of GBM Cells

### 2.1. CD47/SIRP Pathway

As mentioned above, the "don't eat me" signal protein CD47 has emerged as an authentic immune checkpoint target and humanized anti-CD47 monoclonal antibodies are currently in clinical trials [63]. Preliminary data from preclinical mouse models have shown anti-CD47 therapy to be very effective in treating breast, ovarian, and GBM cancers [58,62,88]. Anti-CD47 treatment of mice harboring orthotopically injected luciferase-expressing GBM xenografts resulted in an almost complete inhibition of tumor growth as measured by bioluminescence photon flux [58]. An added benefit of anti-CD47 is that it not only promotes phagocytosis but can promote M1 macrophage polarization which should promote a more immunologically "hot" tumor [89].

Preclinical studies have also focused on the ability of anti-CD47 to work in combination with other therapy. Anti-CD47 can synergize with IR or TMZ to prolong the survival of human GBM xenograft-implanted mice [90]. A recent study showed that TMZ was important for anti-CD47 induced phagocytosis of GBM and this was dependent on the Stimulator of Interferon Genes (STING) [91]. Another study simultaneously blocked VEGF and CD47 using a novel bispecific fusion protein VEGFR1D2-SIRPαD1 [92]. This was effective at both enhancing phagocytosis as well as lowering blood vessel density and angiogenesis. This study also noted that preventing autophagy using chloroquine was able to further enhance these effects. The delivery of therapeutics has always been a challenge for GBM due in part to the blood–brain barrier (BBB). Another laboratory employed an oncolytic virus which encodes an anti-CD47 antibody to improve survival in orthotopic human GBM models [93]. Several groups have been using the implantation of hydrogel formulations into the cavity of freshly resected GBM in rodent brains [94,95]. Song et al. injected hydrogels infused with TMZ and vectors encoding shRNA targeting CD47 [94]. In another study, the use of a temperature-sensitive hydrogel system hydroxypropyl chitin (HPCH) copolymer which encapsulates both anti-CD47 and TMZ was able to provide a full curative effect in approximately 50% of the animals harboring GL261 murine glioma tumors [95].

As mentioned above, interfering with CD47 might have side effects as this protein is expressed on other cell types, most notably red blood cells [96]. Much of the clinical

application of anti-CD47 therapy has been tested in hematological malignancies, in particular AML. Serious side effects have been reported in these recent clinical trials with the anti-CD47 monoclonal antibody Magrolimab [97]. Alternative approaches to interfere with CD47 function involve the use of a SIRPa-Fc fusion protein to bind to and prevent tumor CD47 from engaging SIRP receptors on TAMs [98–100]. Using SIRPa-Fc in an immunocompetent mouse model of GBM was effective in reducing tumor size, particularly when used in combination with autophagy inhibitors [71]. Interestingly, the SIRPa protein has also been detected in normal astrocytes and lower grade glioma; however, its expression decreases with increasing tumor grade [72,73]. Aggregation of the U373MG cell line resulted in an increase in tyrosine phosphorylation of SIRP which was attenuated with an anti-CD47 blocking antibody. It is unclear what role tumor-expressed SIRP proteins play in carcinogenesis or immune evasion.

As alluded to earlier, therapeutic approaches which convert the GBM from an immunologically "cold" to a "hot" are conducted with the aim of rendering the tumor more amenable to immune checkpoint therapies [74,75]. Using TLR3 and TLR9 agonists, Huang et al. were able to stimulate microglia phagocytosis and clearance of GBM cells in tissue culture, brain slices, and in the GL261 mouse model [76]. Another study used "trojan horse" nanoparticles to deliver anti-CD47 along with a STING agonist into the brains of GBM-bearing mice. It was observed in this study that the polarization of TAMs within the tumor was strongly towards the M1 phenotype [77].

Pathways that regulate CD47 expression on tumor cells have gained much attention in recent years. Epigenetic regulation of this phagocytic immune checkpoint seems to play a major role. Tacedinaline (CI-994), a class I Histone Deacetylase (HDAC) inhibitor was identified in a screen as an apoptosis inducer in MYC-driven cancer cell lines including GBM lines [101]. CI-994 treatment significantly enhanced the expression of the "eat me" signals calreticulin and HMGB1 on the surface of the medulloblastoma cell lines tested. The administration of CI-994 combined with anti-CD47 was effective in treating medulloblastoma in orthotopically injected mice. In macrophages, HDAC activity appears to inhibit phagocytosis [102]. The expression of miRNA22 in TAMs represses HDAC6 expression, which in turn results in a higher level of phagocytosis of GBM cell lines. These studies suggest that HDAC inhibition could be effective with checkpoint inhibitor therapy, as it activates pro-phagocytosis pathways in both GBM and TAMs. Another recent study demonstrates a role for the 5 methyl cytosine (5mC) RNA modification of chromatin associated CTNNB1 RNA by the NSUN methyltransferase complex [103]. This modification results in a downregulation of the inhibition of beta-catenin and CD47/SIRP signaling and thus an enhancement of phagocytosis.

In addition to its ability to engage and activate SIRP receptors on TAMs, CD47 plays cell-autonomous roles in GBM [104,105]. The matricellular protein Thrombospondin-1 (TSP-1) can also activate CD47 [106]. The activation of CD47 using a TSP-1 derived peptide agonist increased the proliferation of U87 and U373 cells but not normal human astrocytes [104]. CD47 mediates invasion via the Phosphoinositol-3-Kinase (PI3K) pathway in the U87 and T98G GBM lines [105]. It was also observed that the ablation of CD47 in mouse and human GBM cells resulted in an increase in notch pathway signaling and concomitant upregulation of the extracellular matrix protein Tenascin-C [107]. This increase in Tenascin-C expression resulted in a higher level of TAM infiltration and phagocytosis. Work with glioma stem cells (GSCs) has elucidated other mechanisms of CD47 regulation in this population of cells. It is known that the irradiation of GBM is rarely effective, as GSCs are radioresistant and have the ability to recreate the tumor within a short period of time [108]. It has been demonstrated that GSCs express higher levels of CD47 [109]. This increased CD47 expression seems to contribute to GSC proliferation and migration, as treatment with anti-CD47 slows each of these processes. In a recent paper, it was shown that IR induces CD47 expression on GSCs via the 5′ AMP-activated protein kinase (AMPK) pathway [110]. GSCs also typically display altered metabolic pathways relative to the bulk tumor. For example, GSCs utilize fatty acid oxidation (FAO) to a much greater extent than

bulk GBM tumor cells [111]. In a study by Jiang et al., the FAO pathway resulted in the activation of NF-κB which in turn upregulated CD47 expression specifically in GSCs [112]. The blockade of FAO using etoximir, an inhibitor of a key enzyme in the FAO pathway, synergized with anti-CD47 to strongly increase GBM phagocytosis and decrease tumor volume.

Some insight has been gained in understanding the molecular pathways which regulate CD47 expression and stability on GBM cells. The Leucine-Rich Repeats and Ig-Like Domain (LRIG) family of transmembrane receptors has been associated with neurological tumors [113,114]. There are three LRIG genes in humans and they primarily regulate growth factor receptors by targeting them for ubiquitination. In glioma cells, LRIG1 and 3 act as tumor suppressors, while LRIG2 has a seemingly oncogenic function [113,115,116]. A recent study by Hu et al. showed that LRIG2 can strongly prevent the phagocytosis of GBM cells by upregulating components of the CD47 pathway [117]. In this report, soluble LRIG2 generated by A disintegrin and metalloprotease 17 (ADAM17)-mediated cleavage from GBM cells was shown to recruit TAMs and induce SIRPa expression on their surface. In addition to this, LRIG2 was shown to enhance the expression of CD47 on GBM cells via transcriptional activation of the CD47 gene. This study strongly implicates LRIG2 as a key regulator of the CD47 immune checkpoint via the upregulation of both CD47 and SIRPa. The well-characterized beta-catenin oncogenic pathway can also stimulate CD47 expression in GBM cells [118]. Another study demonstrates how the EGFR pathway cooperates with CD47 to promote tumor progression. EGFR is commonly altered in GBM, although single agent therapy with EGFR inhibitors have displayed modest efficacy at best [119]. In a recent study, EGF treatment or expression of the constitutively active truncation mutant EGFRvIII was able to increase the expression of CD47 in multiple established human GBM cell lines [120]. This effect was found to be mediated by SRC phosphorylation of CD47 at Y288 which prevents binding and polyubiquitination by Tripartite motif-containing protein 21 (TRIM21), an E3 ubiquitin ligase. GBM cells expressing a CD47 Y288F mutant are phagocytosed in vivo at much higher levels, and animals orthotopically injected with these cells survive longer than wild-type injected counterparts.

### 2.2. CSF-1/CSF-1R Pathway

The macrophage growth factor Colony Stimulating Factor-1 (CSF-1) is strongly pro-tumorigenic in most cancers, as the activation of the Colony Stimulating Factor-1 Receptor (CSF-1R) promotes the M2 polarization of TAMs [121–124]. It was originally observed that breast carcinoma invasion and metastasis is severely diminished in CSF-1 null (op/op) background mice [125]. Since that discovery, CSF-1 was shown to be associated with many malignant cancers and the establishment of a generally immunosuppressive microenvironment [126]. Our laboratory has shown that glioma cells express CSF-1 which results in TAM recruitment and the invasion of tumor cells into normal brain parenchyma [23,127,128]. The blockade of CSF-1R signaling with PLX-3397, a CSF-1R pharmacological inhibitor that crosses the blood–brain barrier, strongly attenuated invasion and was curative in some preclinical GBM models [23,129]. In the context of CSF-1R contribution to the phagocytic checkpoint, the balance of evidence points to an important role for CSF-1R signaling to rearrange actin during the various stages of phagocytosis [130]. Recently, it was demonstrated that GW2580, another pharmacological inhibitor of CSF-1R, induced MHC-II expression, phagocytosis, and the T-cell-mediated killing of GBM tumors in patient-derived 3D organoid models [131]. Interestingly, PLX3397 did not display these features. This discrepancy is likely because these inhibitors hit different off-target RTKs [132,133]. Under standard conditions, CSF-1 is released by cancer cells as a soluble ligand which acts on macrophages via the CSF-1R in a paracrine and possibly endocrine fashion [134]. However, it was discovered that there is a splice variant of the CSF-1 gene which contains a transmembrane domain and results in a surface membrane bound form of CSF-1, referred to as mM-CSF [78]. Interestingly, glioma cells expressing mM-CSF are highly susceptible to phagocytosis and cytotoxic killing by TAMs [78–80]. The mechanism is unclear; however,

the membrane-bound mM-CSF is functional as it has the ability to stimulate macrophage colony formation [78]. Therefore, a potential therapeutic approach might be to promote alternative splicing of the CSF-1 gene in GBM tumors towards the membrane-bound isoform.

### 2.3. BACE-1

A recent intriguing study identified the β-site amyloid precursor protein cleaving enzyme 1 (BACE1) as playing a role in preventing tumor phagocytosis. BACE1 is responsible for generating β-amyloid peptides which form plaques in the brains of Alzheimer's disease (AD) patients resulting in the neuronal dysfunction associated with that disease [135]. Several groups have developed BACE-1-specific inhibitors for potential AD therapy. In a paper by Zhai et al., a screen for the phagocytosis of fluorescently labeled human patient–derived GSCs was conducted using a library of compounds which have good blood–brain barrier permeability and low toxicity [81]. The BACE-1-inhibitor MK-8931 was found to enhance phagocytosis. Furthermore, it was demonstrated that BACE-1 is expressed in tumor-promoting macrophages and the administration of MK-8931 along with IR was able to suppress malignant GBM growth in vivo. Interestingly, the oligomers of beta amyloid protein that are associated with AD can stimulate microglia phagocytotic activity against glioma cells [82]. As with many neurodegenerative disorders, microglia dysfunction seems to be a central feature of both gliomagenesis and AD [136].

### 2.4. TREM2

Several very recent exciting yet controversial studies have focused on the potential role of Triggering Receptor Expressed on Myeloid Cells 2 (TREM2) in regulating phagocytosis in GBM [137,138]. TREM2 is expressed primarily on microglial cells and can be activated by a variety of ligands including bacteria, polyanionic molecules, lipoproteins, and nucleic acids [139]. TREM2 signals through DNAX Activator Proteins 10 and 12 (DAP10 and DAP12). DAP10 and DAP 12 can activate several downstream pathways including PI3K/AKT and SYK, a tyrosine kinase which is involved in phagocytosis, particularly at the engulfment stage [140,141]. Peshoff et al. discovered that TREM2+ macrophages are increased in IDH wild-type GBMs and its expression is correlated with phagocytosis markers [137]. Unlike other cancer models where TREM2 knockout slows tumor growth, there was no effect of TREM2 ablation on GL261 or CT-2A glioma progression in vivo syngeneic models. Similarly, Zheng et al. showed a higher level of TREM2+ cells in GBM tumors and a strong association with phagocytosis; however, TREM2 deficiency did not have a beneficial effect on mouse survival using the GL261 model system. However, Sun et al. demonstrated with different GBM cell lines (SB28 and NPA-C54B) that the blockade of TREM2 increased IFNg expression, slowed tumor growth, and enhanced animal survival [83]. Furthermore, this was accomplished using three different methods of TREM2 depletion including using TREM2 knockout mice, TREM2 antisense oligo (ASO) delivery, and TREM2 blocking antibody administration. Whether or not this discrepancy is due to the different GBM cell lines used or some other subtle difference between the model systems tested remains to be seen.

### 2.5. Siglecs

The pattern of glycosylated cell surface proteins and lipids is likely a strategy employed by vertebrates to distinguish self vs. non-self [142,143]. The sialic acid-binding immunoglobulin type lectins (siglec) family of proteins are cell surface receptors expressed on most white blood cells that play a role in immune recognition and regulation [144]. The extracellular region of siglec proteins contains a V-Set Ig domain which binds to sialic acid which is expressed on many glycoproteins and glycolipids. There are 14 human siglecs and 9 murine siglecs; they are thought to play an important role in immune checkpoints in several diseases. Siglecs often act as coreceptors and can activate or inhibit immune responses depending on their C-terminal domain structure. Siglecs can modulate several

cellular functions including endocytosis, antigen presentation, and phagocytosis. Most siglecs contain immunosuppressive ITIM and immunoreceptor tyrosine-based switch motif (ITSM) motifs and are therefore immunosuppressive. However, a subset of siglecs do not contain ITIMs and instead have a region which can associate with the adaptor DAP12 which as noted above is involved in immune activation and phagocytosis. The hyperglycolysation of tumor cells is associated with a poor outcome [145,146]. Certain siglecs are expressed primarily on macrophages and microglia. These include Siglec-7, 9, 10, and 15, and they have been the subject of investigation for immune checkpoint therapy [147]. For example, the expression of Siglec-E in TAMs was found to inhibit tumor growth [148]. Furthermore, Siglec-10 engagement of CD24 on breast carcinoma cells inhibited phagocytosis [149]. In the context of GBM, Siglec-H expression is increased in IFNg-treated M1-polarized microglia [84]. Interestingly, Siglec-H only binds to murine GBM cells and not normal mouse cells, highlighting the hyperglycosylated aspect of cancer cells. This interaction resulted in the phagocytosis of the murine GBM cell lines SMA560 and GL261 and this was dependent on DAP12. Additional siglec family members act as checkpoints on TAMS. Siglec-1 (also called CD169/Sialoadhesin) is expressed predominantly on macrophages and is seemingly the only member of the siglec family which does not initiate either pro- or anti-immune signaling [85,86]. The role of Siglec-1/CD169 was recently investigated in a GBM model [67]. The depletion of CD169 resulted in a decrease in CXCL10 expression and T-cell infiltration. CD169 also played an important role in the phagocytosis of GBM cells. Another siglec that is mainly expressed on macrophages is Siglec-15. The role of Siglec-15 was investigated in a recent paper where it was discovered to be expressed primarily by peri-tumoral macrophages within GBM tumors [87]. The pattern of expression of Siglec-15 changes during the course of GBM progression as it reaches its highest expression in grade II astrocytoma. A negative correlation between Siglec-15 expression and CD3+ cell infiltration was noted. The knockout of Siglec-15 in a mouse macrophage cell line increased M1 marker expression, decreased M2 marker expression, and enhanced the phagocytosis of GL261 glioma cells.

Strategies developed thus far towards these ends have mainly focused on using siglecs as direct targets on tumor cells [150]. The receptor CD22 (Siglec-2) has received a lot of attention as it is expressed on many B-cell lymphomas. Monoclonal antibody drug conjugates against Siglec-2 have been developed and have been approved for treating acute lymphoblastic leukemia and hairy cell leukemia. Anti-CD22 has also been used to generate CAR-T therapy for these leukemias and these treatments are undergoing clinical trials [151]. Anti-CD33 (Siglec-3) monoclonal antibodies and CAR-T have been generated as a therapy for myeloid leukemias [152,153]. Siglecs also represent bona fide immune checkpoints that can be targeted to enhance the immune response [147,154]. The removal of sialic acids from the surface of tumor cells is an alternative strategy. For example, Palleon Pharmaceuticals has developed human sialidase-coupled antibodies which are currently in phase I/II clinical trials.

### 2.6. Phagocytosis Checkpoint Cooperation with Anti-CTLA4/PD-1/PD-L1 ICIs

There is some evidence that the immune checkpoint inhibitors currently approved for clinical use to reverse T-cell inhibition may promote phagocytosis pathways. For example, IR and anti-PD-L1 treatment resulted in an increase in T-cell and TAM recruitment to the tumor in murine genetic models of GBM [155]. Anti-PD-L1 was shown to strongly enhance macrophage phagocytosis of GBM cells and this was independent of T cells. Chen et al. recently discovered that using anti-CTLA4 antibodies resulted in an increase in IFNg release within GBM tumors which stimulated DC and microglia phagocytosis of tumor cells [156]. Phagocytic activity in this context was dependent on signaling from Axl and Mer RTKs.

## 2.7. Chimeric Antigen Receptor-M

One of the approaches of modern immunotherapy involves ex vivo manipulation of a patient's peripheral immune cells to be anti-tumorigenic followed by reintroduction into the patient [157,158]. Most work in this area has been with cytotoxic T cells. In particular, researchers have generated vectors which express chimeric antigen receptors whereby the ectodomain of the T cell receptor is replaced with a domain that specifically and directly interacts with target receptors on the surface of cancer cells [159]. In this way, the antigen presentation steps which involve multiple immune checkpoint coreceptors (discussed above) is completely bypassed. CAR-T has been quite effective for several cancers, especially leukemias [160,161]. Unfortunately, as with other immunotherapies, CAR-T has not delivered measurable clinical responses with GBM [162]. Other leukocytes are now being tested for CAR-style therapy, including macrophages. CAR-macrophages (CAR-Ms) are currently receiving a lot of attention from researchers and biopharmaceutical companies [163]. Unlike T cells, which often cannot penetrate solid tumors very well, macrophages are quite good at tumor infiltration [164]. One of the major strategies for CAR-M therapy is to prevent "don't eat me" signals, such as CD47, to be conveyed to the macrophage [165]. There is some evidence in animal models that M1 polarization can improve CAR-M therapy [166]. Macrophages transduced with an HER2-Fc-Receptor chimeric antigen receptor treated with the M1 polarizing stimuli IFNg and LPS were more effective than unstimulated controls in a breast carcinoma model. Finally, purposing CAR-M to specifically target GSCs might be an effective approach for eradicating these cells as a recent paper has indicated [167]. There are, however, limitations to CAR-M therapy. Severe toxicity due to a "cytokine storm" remains a serious potential issue [168]. Furthermore, the complexity of macrophage biology and interaction with tumor cells and other immune cells within the microenvironment must not be underestimated [164,169,170]. As this technology becomes more advanced and widespread, opportunities for targeting other checkpoint targets will be exploited.

## 3. Conclusions and Perspectives

We are in an exciting era of cancer therapy as gains are being made (albeit slowly) in treating metastatic cancers. Much work remains to be conducted, but there are clear paths delineated to the discovery of multiple immune checkpoints. GBM treatment will likely require a multi-pronged approach utilizing chemotherapy to induce GBM cell death in combination with treatments that inhibit M2 and promote M1 macrophage polarity and block phagocytosis immune checkpoints. The pathways and mechanisms that regulate the phagocytosis checkpoints at the glioma cell/TAM interface are summarized in Figure 1. LRIG2 is an enticing target as it upregulates CD47 and SIRPa expression in GBM cells and TAMs, respectively [117]. It may also be time to revisit EGFR inhibitors for GBM treatment as this pathway enhances CD47 stability in addition to its other oncogenic activities. The "cancer glycome" also presents itself as a target-rich environment for immune checkpoint blockade therapy [171,172]. Finally, to fully eradicate GBM tumors, the GSC population must be effectively eliminated. The discovery that GSC cells are more dependent on CD47 expression could greatly assist in developing treatments to eliminate these cells.

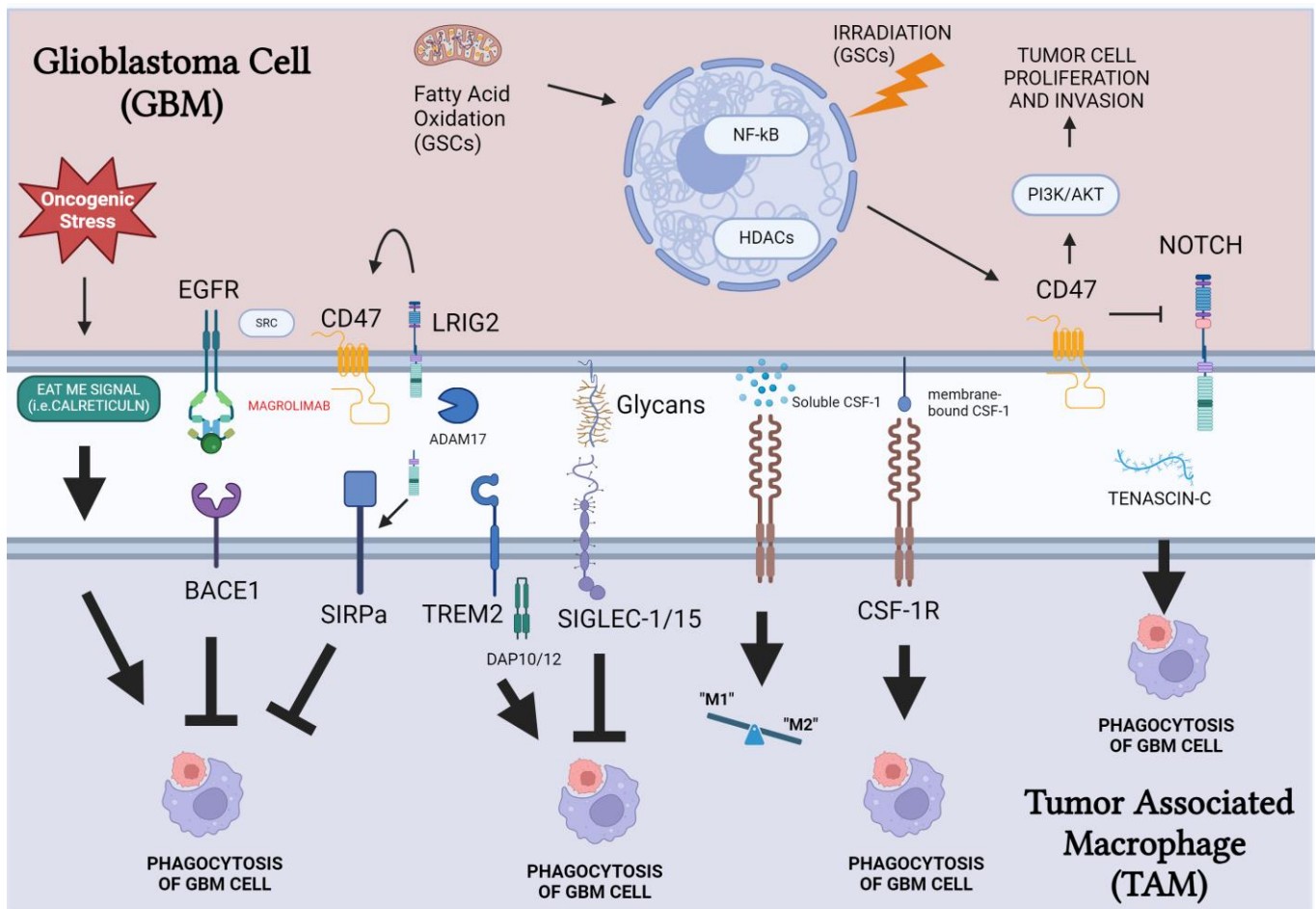

**Figure 1.** Phagocytosis checkpoints at the glioma/macrophage interface. CD47 exhibits bidirectional signaling that promotes both GBM tumor invasion and inhibition of phagocytosis by engaging SIRP receptors on the surface of TAMs. Pathways which regulate CD47 expression include EGFR/SRC, LRIG2, and NF-κB. TREM2 can also promote phagocytosis, while siglecs tend to be strongly immunosuppressive and represent another phagocytosis checkpoint along with CD47. Effective therapy for GBM will likely involve a combination of treatments which induce GBM cell death, promote an immunologically "hot" tumor by activating M1 macrophage pathways and stimulation of GBM phagocytosis by TAMs using checkpoint inhibitors.

**Author Contributions:** Conceptualization, S.J.C.; Investigation, A.A., Z.A., S.B., A.D., S.M., Y.M. and S.J.C.; writing—S.J.C.; writing—review and editing, A.A., Z.A., A.D., S.B., S.M. and Y.M. All authors have read and agreed to the published version of the manuscript.

**Funding:** This research received no external funding.

**Data Availability Statement:** Data are contained within the article.

**Acknowledgments:** Thanks to Kean University School of Integrative Science and Technology and Department of Biological Sciences.

**Conflicts of Interest:** The authors declare no conflicts of interest.

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
