# Peer review of "Phagocytosis Checkpoints in Glioblastoma: CD47 and Beyond"

_cimb, doi:10.3390/cimb46080462_

Round 1

Reviewer 1 Report

Comments and Suggestions for Authors

This manuscript, entitled Phagocytosis Checkpoints in Glioblastoma: CD47 and Beyond, discusses the role of immune checkpoint inhibitors, specifically CD47, in glioblastoma multiforme (GBM), a highly aggressive and drug-resistant extremely aggressive and drug-resistant brain cancer. The manuscript emphasizes the importance of understanding the mechanisms by which macrophages phagocytose GBM cells and the potential for therapeutic intervention by targeting CD47 and other immune checkpoint molecules. This is a topic of great clinical relevance, but the manuscript still has significant shortcomings and requires further revision and refinement.

1. The manuscript provides a comprehensive overview of glioblastoma multiforme (GBM) and its therapeutic challenges. However, it would benefit from a more clearly structured and concisely worded introduction. The discussion of the limitations of standard treatments and the transition with the presentation of the latest immunotherapies could have been smoother. In addition, briefly mentioning the points that will be covered in subsequent chapters, providing the reader with a roadmap, and adding corresponding tables detailing the findings of different studies would have added clarity to the article.

2 The manuscript effectively explains the role of immune checkpoint inhibitors (ICIs) and the limited efficacy of ICIs in GBM due to the immune "cold" nature of these tumors. However, this section could be strengthened by providing more specific examples of studies exploring ICIs in GBM, including any promising results or ongoing clinical trials, and this section should incorporate more recent literature and reports, with the addition of corresponding tables to better detail this section.

3. The discussion on CD47 as an immune checkpoint is very detailed, especially the explanation of "don't eat me" signaling and its interaction with SIRPα. However, this section could have been strengthened by the inclusion of more recent studies focusing on the therapeutic potential of targeting CD47 in GBM, especially recent findings from the last 3 years.

4. Other pathways involved in phagocytosis by macrophages (e.g., TREM2 and sugar-binding proteins) have been elaborated in this study. However, the manuscript would have benefited from a more in-depth exploration of how to target these pathways therapeutically. Incorporating data from preclinical or clinical studies examining the role of these targets in GBM would provide stronger support for their potential as therapeutic strategies.

5. This section also allows for further discussion of the challenges and limitations of CAR-M therapies, such as potential off-target effects, manufacturing complexities, and the current status of CAR-M therapies in clinical development. Providing a balanced perspective will help readers understand the potential and obstacles of this emerging therapeutic approach.

6. The English presentation and language of the manuscript require further refinement and modification in order to present the relevant content more clearly.

Comments on the Quality of English Language

The English presentation and language of the manuscript require further refinement and modification in order to present the relevant content more clearly.

Author Response

We thank the reviewer for very helpful and insightful comments which have greatly enhanced the quality of the manuscript. We addressed each of the concerns below:

Comment 1. The manuscript provides a comprehensive overview of glioblastoma multiforme (GBM) and its therapeutic challenges. However, it would benefit from a more clearly structured and concisely worded introduction. The discussion of the limitations of standard treatments and the transition with the presentation of the latest immunotherapies could have been smoother. In addition, briefly mentioning the points that will be covered in subsequent chapters, providing the reader with a roadmap, and adding corresponding tables detailing the findings of different studies would have added clarity to the article.

Answer: We have expanded the introduction to include more background on GBM in general and provided much more detail about ICIs and recent clinical trials with GBM. We have also included a table at the end of the introduction to serve as a overall guide to the review.

Comment 2 The manuscript effectively explains the role of immune checkpoint inhibitors (ICIs) and the limited efficacy of ICIs in GBM due to the immune "cold" nature of these tumors. However, this section could be strengthened by providing more specific examples of studies exploring ICIs in GBM, including any promising results or ongoing clinical trials, and this section should incorporate more recent literature and reports, with the addition of corresponding tables to better detail this section.

Answer: The introduction now contains a comprehensive description of the clinical trials performed with ICIs in GBM patients. 

Comment 3. The discussion on CD47 as an immune checkpoint is very detailed, especially the explanation of "don't eat me" signaling and its interaction with SIRPα. However, this section could have been strengthened by the inclusion of more recent studies focusing on the therapeutic potential of targeting CD47 in GBM, especially recent findings from the last 3 years.

Answer 3: We have included all relevant CD47 data with respect to GBM. In particular we added additional studies that were carried out over the last three years.

Comment 4. Other pathways involved in phagocytosis by macrophages (e.g., TREM2 and sugar-binding proteins) have been elaborated in this study. However, the manuscript would have benefited from a more in-depth exploration of how to target these pathways therapeutically. Incorporating data from preclinical or clinical studies examining the role of these targets in GBM would provide stronger support for their potential as therapeutic strategies.

Answer 4: We added an additional paragraph covering the potential of Siglecs as targets for ICI and relevant clinical trials.

Comment 5. This section also allows for further discussion of the challenges and limitations of CAR-M therapies, such as potential off-target effects, manufacturing complexities, and the current status of CAR-M therapies in clinical development. Providing a balanced perspective will help readers understand the potential and obstacles of this emerging therapeutic approach.

Answer 5: We included this information on potential downsides and risks of CAR-M in order to balance the perspective in this section.

Comment 6. The English presentation and language of the manuscript require further refinement and modification in order to present the relevant content more clearly.

Answer 6: Manuscript has been proofread to fix grammatical and spelling errors.

Reviewer 2 Report

Comments and Suggestions for Authors

The authors have concisely reviewed the mechanisms and signaling of CD47 as phagocytosis checkpoints in Glioblastoma, with a good diagram to illustrate the pathways involved.

The authors may describe more about Glioblastoma multiforme (GBM) in the introduction, such as how common is the diseases. Also, the research and clinical progress of using TAM phagocytosis in cancer immunotherapy.

In the conclusion, the authors may add some clinical perspectives on CD47, and the pros and cons of CD47 as an immunotherapy target.

Author Response

We thank the reviewer for very helpful and insightful comments to improve the manuscript.

Comment 1: The authors may describe more about Glioblastoma multiforme (GBM) in the introduction, such as how common is the diseases. Also, the research and clinical progress of using TAM phagocytosis in cancer immunotherapy.

Answer 1: We greatly expanded the introduction to include this information as well as relevant clinical trials with ICI therapy and GBM.

Comment 2:In the conclusion, the authors may add some clinical perspectives on CD47, and the pros and cons of CD47 as an immunotherapy target.

Answer 2: We included more recent studies with CD47 therapy and added a reference to clinical trials in AML conducted with Magrolimab which discusses toxic side effects of this drug.

Round 2

Reviewer 1 Report

Comments and Suggestions for Authors

The authors revised the manuscript line by line, and at this stage I think the revisions were sufficient.

Comments on the Quality of English Language

English expression has been improved and refined from the previous version.